# Using a Pair of Different-Sized Spheres to Calibrate Radar Data in the Microwave Anechoic Chamber

**Jiangkun Gong** [1] , **Jun Yan** [1] , **Deren Li** [1] , **Huiping Hu** [2] and **Deyong Kong** [3,*]

1   State Key Laboratory of Information Engineering in Surveying, Mapping and Remote Sensing, Wuhan University, Wuhan 430079, China
2   Wuhan Geomatics Institute, Wuhan 430022, China
3   School of Information Engineering, Hubei University of Economics, Wuhan 430205, China
*   Correspondence: kdykong@hbue.edu.cn; Tel.: +86-027-6877-8527

**Featured Application: The calibration method using a pair of different-sized spheres to calibrate radar data in the anechoic chamber can be used in applications widely, such as measuring radar cross-section (RCS) and radar signatures of objects.**

**Abstract:** Traditional radar calibration methods often use one standard sphere as the standard calibration object, but researchers seldom discuss the size and the material content of the calibration sphere. We propose using a pair of different-sized spheres to verify the standard method. In a microwave anechoic chamber, the simulated and measured results show that the radar cross-section (RCS) values of two spheres with radii of 0.15 m and 0.05 m in the Mie region differ from their physical cross-sections in the optic region, and only in the optic region, their difference of RCS value (i.e., 9.5424 dB) can be approximately equal to the theoretical one (i.e., 9.6803 dB). Thus, radar calibration should be conducted in the same scattering region for both the calibration object and the calibrated target. The use of two different-sized spheres can aid in three applications: (1) verifying the scattering regions, (2) searching the pure area in the microwave anechoic chamber, and (3) locating the positions of the targets in the range profiles.

**Keywords:** different-sized spheres; microwave anechoic chamber; radar calibration; scattering region

## 1. Introduction

Researchers need to collect and calibrate the radar data of targets to analyze radar signatures. The radar cross-section (RCS) value of a target is the first of many radar signatures that need to be measured. There are several approaches to calibrating the RCS value [1,2]. The most practical method for calibrating the RCS value of a measured target is to use a standard object with a known RCS. This is known as the ratio method [3,4], in which the signal amplitude of the target is compared to that of the calibration object. The process is described by [5]

$$\sigma_m = \frac{P_m}{P_r} \sigma_r \tag{1}$$

where $\sigma_m$ is the RCS value of the measured target, $\sigma_r$ is the known RCS value of the standard object, $P_r$ is the received power of the standard object, and $P_m$ is the received power of the measured target. The isotropic conducting sphere is the most commonly used calibration object because its RCS value within different scattering regions can be calculated theoretically using different scattering rules [6]. For example, its RCS in the Rayleigh region and Mie region can be calculated using complex equations with the Rayleigh scattering principle and Mie scattering rules, respectively. However, in the optic region, the RCS value ($\sigma_s$) of an isotropic sphere with a diameter of $r$ can be determined based on its geometric cross-section [6,7], which is

$$\sigma_s = \pi r^2 \tag{2}$$

Note that Equation (2) is only effective in the optic region. However, many available measurements seldom consider this condition, and researchers use incorrectly sized spheres to calibrate the radar data within different scattering regions. For example, Samuel S. Urmy et al. noted that the RCS value of a stainless-steel sphere with a diameter of 20.3 cm as 0.0321 m$^2$ should be calculated by the Mie series solution, but they still used this value to calibrate an X-band radar to detect birds, whose radar data were in the optic region [5]. They seemed to ignore the scattering mechanism within the Mie region, as well as the material of the sphere and its influence on calibrating radar echoes from birds [5]. Some previous work used two different standard spheres with known RCS values [4]; however, they did not discuss the relationship between the radar calibration and the scattering region.

In this paper, we propose a method to calibrate the standard method of one standard sphere using a pair of different-sized spheres, and we emphasize that radar calibration should consider the scattering regions when selecting the calibration sphere. The method provides references for answering questions concerning how big the standard sphere is, the material of the standard sphere, and what the proper distance between the standard sphere and the antenna should be. In Section 2, we present the method using two spheres with radii of 0.05 m and 0.15 m. In Section 3, we analyze the results using the radar data of the two spheres in various radar bands, including the S-band (2~4 GHz), X-band (8~12 GHz), and Ku-band (14~18 GHz), in an anechoic chamber to support our claim. In Section 4, we discuss the results, describe the procedure of the method using a pair of different-sized spheres, and present the potential applications. Finally, in Section 5, after summarizing the method, the results, and the applications, we conclude that our method can be a practical approach to validate the standard radar calibration method using one sphere.

## 2. Materials and Methods

The RCS value of a target can be measured in an anechoic chamber or the outfield. In the outfield, the measuring tools are real radar systems [5], while in an anechoic chamber, a vector network analyzer (VNA) acts as the "radar" system [3]. Network analyzers are convenient tools for onsite RCS measurements [8]. They measure the S-parameter matrix of a system. The transmitting antenna is connected to source port 1 of the network analyzer; the receiving antenna is connected to source port 2. The parameter S21 (unit: dB) describes the ratio of the transmitted power to the received power. This value represents radar echoes from the measured targets in the anechoic chamber. Thus, based on S21, the Equation (1) is given as

$$\sigma_m{}' = S_m - S_r + \sigma_r{}' \tag{3}$$

where $\sigma_m{}'$ is the RCS value of the measured target (unit: dBm$^2$), $\sigma_r{}'$ is the known RCS value of the standard object (unit: dBm$^2$), $S_r$ is the received power of the standard object (unit: dB), and $S_m$ is the received power of the measured target (unit: dB).

We conducted a serial measurement in an anechoic chamber in 2019. The anechoic chamber belongs to Marine Equipment Inspection & Testing in Qingdao, China (http://www.chinanmei.com/, accessed on 10 June 2022). The inner room of the anechoic chamber is 7.890 × 3.517 × 3.3 m, as shown in Figure 1. The antennas are double-ridged broadband horn antennas, BBHA 9120 D, manufactured by Schwartzbeck from Schönau, Germany. The size of an antenna is 245 × 142 × 408 mm. The detailed VNA used in this analysis was PNA-N5224A, which was manufactured by Keysight Technologies (Santa Rosa, CA, USA). Figure 1b shows the photo of the internal environment in the microwave anechoic chamber. The wedges are wave absorption wedges. We used the swept frequency measurement of the VNA to obtain radar data with a high-range resolution profile (HRRP). The VNA antennas used vertical polarization to transmit signals and receive echoes. If the measured band is $B$, the sampling data are $N$, and the parameters can be calculated as

$$f_e = \frac{B}{N} \tag{4}$$

$$t_e = \frac{1}{f_e} \tag{5}$$

$$R_e = \frac{c}{2B} \tag{6}$$

$$L = \frac{ct_e}{2} \tag{7}$$

where $c$ is the propagation velocity of the electromagnetic wave, $f_e$ is the frequency resolution, $t_e$ is the time dynamic range, $R_e$ is the range resolution, and $L$ is the maximum measurement range. The advantage of HRRP technology is that it can separate radar echoes from different target structures [9]. Table 1 shows the setup parameters of the VNA when measuring data in the S-band, X-band, and Ku-band. Figure 2 depicts the data from the background environment when no object was on the foam holder. In this case, the data from everything in the anechoic chamber except the tested object belonged to the background clutter. The horizontal axis on the screenshot of the VNA is time, which was also the measured distance from the network analyzer. The distance must account for the length of the RF (radio frequency) cables that connect the antennas and the network analyzer. The test area of the foam holder was calculated to be approximately 11.5 m based on the data, which indicates that the foam holder was 11.5 m from the VNA, including the cable length.

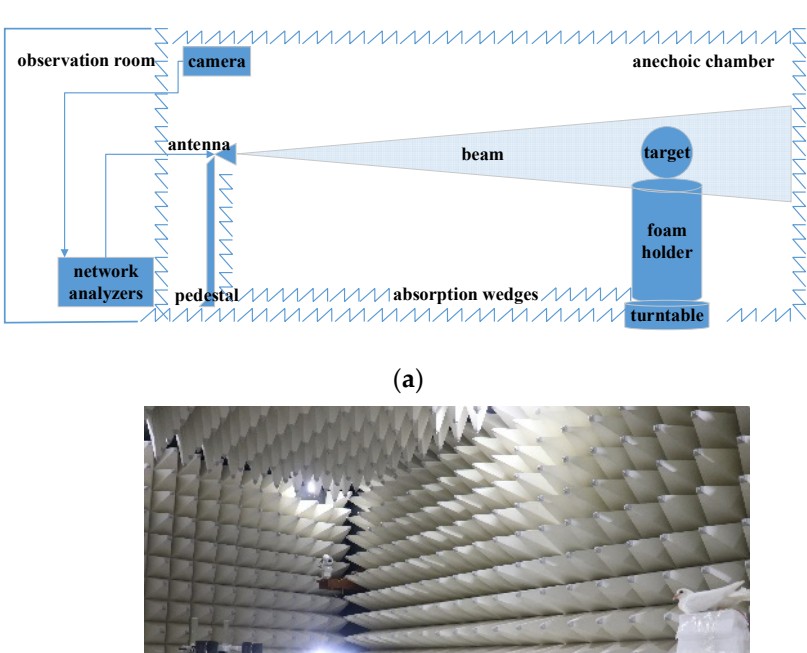

(**a**)

(**b**)

**Figure 1.** Environment of the anechoic chamber: (**a**) the spatial arrangement during the test, (**b**) a photo of the test.

We used two different sized spheres to calibrate the radar data and ensure that the test environment matched the conditions. The radius of the large sphere was 0.15 m, while that of the small sphere was 0.05 m. Theoretically, the RCS values of the two spheres ($\sigma_{a10}$ = 0.05 m, $\sigma_{a30}$ = 0.15 m) in the optic region can be calculated using Equation (2), resulting in values of approximately $-21.0491$ dBm$^2$ and $-11.5067$ dBm$^2$, respectively.

The difference between the RCS values of the two spheres was 9.5424 dB, which was calculated by

$$\sigma_{d} = \sigma_{a30} - \sigma_{a10} \tag{8}$$

We measured the radar data in three radar bands: the S-band (2~4 GHz), the X-band (8~12 GHz), and the Ku-band (14~18 GHz). The S-band had a radar bandwidth of 2 GHz, while the X-band and Ku-band both had radar bandwidth of 4 GHz. The number of sampling points was 1600. The range resolution was 0.075 m for the S-band and 0.0375 m for the X-band and Ku-band. We also measured the radar data at 12 angles ranging from 0° to 360° with a 30° interval in each radar band. The RCS value of any sphere in the Rayleigh region and Mie region is positively correlated with the radar frequency. When the spheres act as calibration objects for each other, the background radar data can be counterbalanced. The primary measurement error was caused by the equipment, including the network analyzer, the pair of antennas, and the cables. We measured for a considerable amount of time to collect the radar data of all of the spheres at each angle and in each radar band. Then, we calculated the mean RCS values of the spheres in the three radar bands.

**Table 1.** Parameters of the network analyzer.

| Content | Parameters | | |
|---|---|---|---|
| **Test Band** | **S** | **X** | **Ku** |
| Frequency span (GHz) | 2~4 | 8~12 | 14~18 |
| Sampling point | 1601 | 1601 | 1601 |
| IF bandwidth (Hz) | 100,000 | 100,000 | 100,000 |
| Power level (dBm) | 12 | 12 | 12 |
| Time span (ns) | 70~85 | 70~85 | 70~85 |

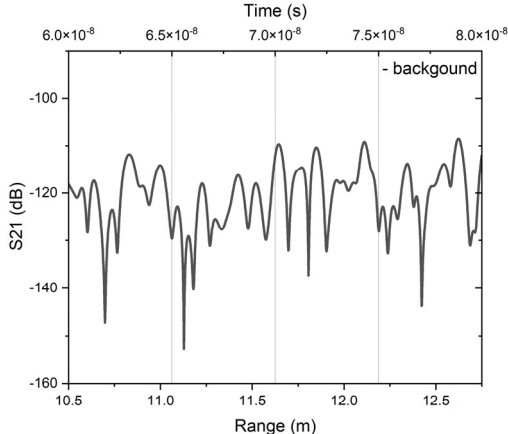

**(a)**

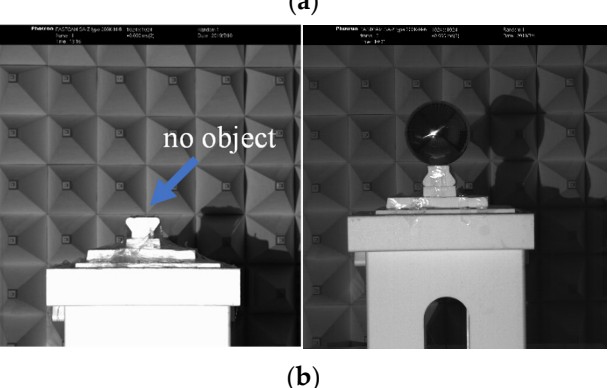

**(b)**

**Figure 2.** Example from the background environment: (**a**) data of the background, (**b**) a photo of no object on the foam holder, and an image of a large sphere on the foam holder.

The RCS value of a sphere depends on the scattering regions. We introduced the electric size, $ka$, to calculate the RCS values of the spheres in the three scattering regions. The electric size, $ka$, of a sphere is given by [6,9]

$$ka = \frac{2\pi a}{\lambda} \tag{9}$$

where $\lambda$ is the radar wavelength and $a$ is the radius of the sphere. In the optic region, when $ka > 10$, the sphere RCS calculated using Equation (2) becomes constant and does not change with the electric size. In the Mie region, when $1 < ka < 10$, the scattering mechanism follows Mie scattering rules, which state that the RCS value of a sphere will be amplified if the frequency is resonant with the natural frequency of the sphere. We chose radii of 0.05 m and 0.15 m because of the rules in the scattering regions. The typical wavelength values of the S-band, X-band, and Ku-band are 10 cm, 3 cm, and 2 cm, respectively, and their radar data in different radar bands are in different scattering regions. The relationships are shown in Table 2.

**Table 2.** Radar bands and sphere sizes.

| Radar Band | Scattering Region | |
|---|---|---|
| | Large Sphere (Radius: 0.15 m) | Small Sphere (Radius: 0.05 m) |
| S-band | Mie and optic region | Mie region |
| X-band | Optic region | Mie and optic region |
| Ku-band | Optic region | Optic region |

## 3. Results

The RCS values of the two spheres in radar bands ranging from 2 GHz to 18 GHz follow the scattering region theory. The simulated values are shown in Figure 3. These results were calculated with the Phased Array System Toolbox in MATLAB 2009b [10]. We found that the curves in the colored frames corresponded to the analysis in Table 2. Only in the Ku-band (14~18 GHz) are the RCS values of the two spheres constant. The scattering data of the small sphere was only in the Mie region in the S-band. Other situations also occurred between the Mie region and the optic region. It should be noted that the term "constant" indicates that the fluctuation of the RCS values and the increasing radar frequency is continuous, and the change is small enough to be considered a statistical error. However, the RCS values of the spheres vibrated with the radar frequency when scattering occurred in the Mie region. The peak values were much larger than those in the optic region. If we calculate the mean amplitude within three radar bandwidths, which is equal to the quotient, when dividing the sampling number of frequency points in one radar bandwidth by the total number of all sampling amplitudes within the bandwidth (e.g., 2 GHz within S-band, 4 GHz within X-band, 4 GHz within Ku-band), the ratio values of that large sphere to the small sphere are 0.5589, 0.5468, and 0.5461, respectively.

Furthermore, one of the resonance effects in the Mie region can be described as that a small frequency deviation can result in large changes in the RCS of a target. For example, it is reported that the scattering resonance effects in echoes from migrating birds are so strong that a 10% frequency deviation within the S-band can result in more than 10 dB changes in the reflectivity values [11]. Thereby, the standard deviation numbers of these data within the S-band, X-band, and Ku-band are 1.823, 0.423, and 0.200 for the small sphere, but 0.520, 0.066, and 0.0223 for the large sphere. In brief, the resonance effect in the Mie region could be the primary explanation for why the RCS values of the two spheres differ from their physical cross-sections.

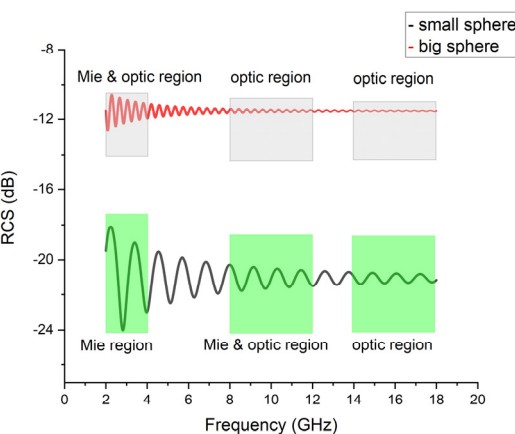

**Figure 3.** Theoretic RCS values of the two spheres in the radar bands (2~18 GHz).

The results of the two spheres in the anechoic chamber also support the above conclusions. Figure 4a–c shows the measured HRRPs of the two spheres in the three radar bands. Since the spheres are isotropic scattering targets, their signal envelopes appeared as a single peak at approximately 11.5 m in the range profile. These ranges represent the "detection range" of the spheres, which includes the length of the wires connecting the antennas to the network analyzer ports and the distance between the antennas and the target; thus, the ranges were longer than the size of the inside room (i.e., 7.9 m). The turntable center was located in the center of the receiving and transmitting antennas, but the signal peaks of both spheres did not remain in the same position at the 12 angles. Three possible reasons may cause this kind of position shift. (1) The turntable center is not exactly in the center line of the two antennas. (2) The directional diagrams of the two antennas may not be the same. (3) The spheres are not perfect because of machining errors.

The width of the signal envelopes of the large sphere was much greater than that of the small sphere. The former was approximately 30 cm, while the latter was 10 cm. These values are equal to the diameters of two spheres. The large sphere had higher signal amplitudes in the three radar bands than the small sphere, and the differences are about 10 dB, as shown in Figure 4a–c. The difference between the maximum and the minimum within the S-band, X-band, and Ku-band are 2.159 dB, 1.475 dB, and 2.034 dB for the large sphere, but they are about 5.112 dB, 10.077 dB, and 2.368 dB for the small sphere, respectively. The data of the large sphere are more stable than those of the small sphere, and the position shifts of the small sphere are more likely to be affected by the radar bands than those of the large sphere.

We used the peaks of envelopes to calibrate the RCS values of the spheres. At each angle, we measured the radar data of each object 20 times. The difference between the maximum and minimum of the 20 samplings differed in the radar bands. The difference in the S-band was the smallest, the difference in the X-band was larger, and the difference in the Ku-band was the largest. However, none of these differences were larger than 0.5 dB. Then, we calculated each peak of the signals in each radar band at each angle and drew the polar diagrams, as shown in Figure 4d. We found that the distribution of the RCS values and grades of the large sphere in the three bands was more uniform than that of the small sphere. The signal amplitude of the small sphere fluctuated in the X-band due to scattering between the Mie and optic regions. Compared with the radar bands, the attitude change has little effect on the signal amplitudes of both the large and small spheres. In Figure 4d, the mean difference between the small sphere and the large sphere was 10.3093 dB within the S-band, 8.5487 dB within the X-band, and 9.6803 dB within the Ku-band. As listed in Table 2, scattering regions are responsible for the difference between the measured values (i.e., 10.3093 dB, 8.5487 dB, and 9.6803 dB) within the three radar bands and the theoretical one (i.e., 9.5424 dB, calculated using Equation (8)). First, both the large sphere and the small sphere are in the optic region (i.e., Ku-band), and their measured difference (i.e., 9.6803 dB)

was closest to the theoretical value of 9.5424 dB among the three bands. Second, when the test was in the X-band, the radar data of the small sphere was amplified due to the resonant effect in the Mie region, but that of the big sphere was not amplified because it stayed in the optic region. Thus, the difference of 8.5487 dB within the X-band was smaller than that of 9.6803 dB in the Ku-band, and is also farther from the theoretical value of 9.5424 dB. Finally, since the S-band radar data of the large sphere and small sphere were in the Mie region, they were both amplified to some degree. As a result, the value of 10.3093 dB was greater than 9.5424 dB. In this case, the resonant effect contributed more to the scattering power of the large sphere than that of the small one. However, in other cases, the value may be closer to 9.5424 dB. In conclusion, when the data of any sphere is in the Mie region, the difference in the RCS values of the two spheres differs from the difference of theoretical geometrical cross-sections.

The rules of scattering regions must be considered in radar calibration. Thus, the relationship between the calibration object and the calibrated target, such as the size and material compositions, must be considered. Table 3 shows the differences in the data between the other objects and the small sphere in other tests. The data of the small sphere were set as 0 dB, and the data of the other objects were relative values to those of the small sphere. The drone was a quad-rotor drone, the DJI Phantom 3, with a general size of about 40 cm. The bird was a seagull with a size of approximately 40 cm. It presents the different data for the drone, the seagull, and the big sphere over that of the small sphere. Based on the ratio of the targets to the spheres, the scattering regions differed for the different targets. According to the scattering region theory, the data in the Ku-band are in the optic region for each object, while all data in the S-band are in the Mie region.

**Table 3.** Relative values of radar scattering power of different objects.

| Objects | S-Band | X-Band | Ku-Band |
|---|---|---|---|
| Small sphere (0.1 m) | 0 dB | 0 dB | 0 dB |
| Big sphere (0.3 m) | 13.162 dB | 13.000 dB | 8.723 dB |
| Seagull (0.4 m) | −6.624 dB | −2.496 dB | −8.155 dB |
| Drone (0.4 m) | 10.987 dB | 7.831 dB | 7.933 dB |

As we stated above, only in the Ku-band, can the radar data of the small sphere be used for calibrating that of the larger sphere. Since the sizes of the seagull and drone are also bigger than the large sphere, the small sphere can be used for calibrating that of the seagull and drone within the Ku-band. Radar calibration in the optic region requires no restrictions on the material composition of the calibration object, but the selected size of the calibration object must ensure that the calibration object's data remain in the optic region. The projected area can be used to calculate the theoretical RCS value of the calibration object. If the calibration object is a sphere, Equation (2) can be used for calculating the RCS value of the sphere.

Calibration in the Mie region is more complex than calibration in the optic region. Different resonance effects occur in the data in the resonance region of targets composed of different materials, and it is hard to quantify the contribution of the resonance effect in the Mie region to the RCS values of objects. As shown in Table 3, the relative numbers of the drones in the Ku-band and S-band were 7.933 dB and 10.987 dB, respectively, but the values of these numbers of the seagull are −8.155 dB and −6.624 dB, respectively. This means that the resonance effect within the S-band contributes more to the bird than the drone. Moreover, since the RCS values of the calibration object (e.g., spheres) were not equal to their physical cross-section in the Mie region, it is not advisable to use the small sphere to calibrate the radar data of the drone and the bird in the Mie region. In conclusion, it is not proper to use the small sphere to calibrate any one object without considering the scattering region, and more importantly, it is not reasonable to choose any sphere (small sphere or large sphere) to calibrate one specific object. The scattering region must be the first rule for radar calibration.

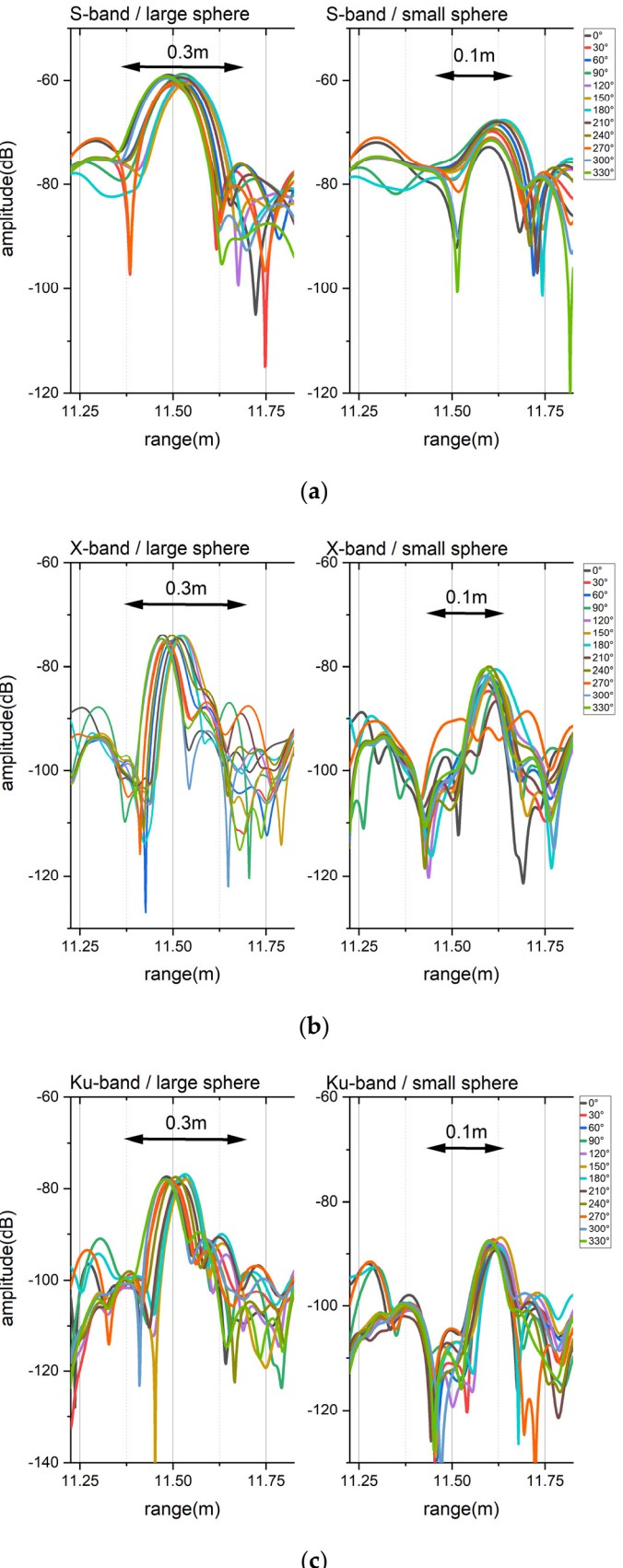

(**a**)

(**b**)

(**c**)

**Figure 4.** *Cont.*

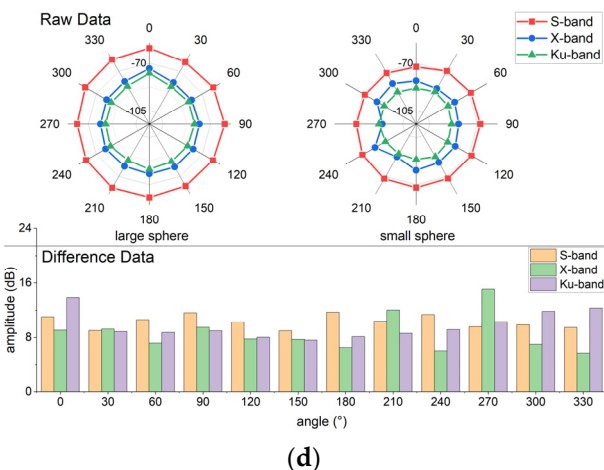

**(d)**

**Figure 4.** Radar data of the two spheres within three radar bands: (**a**) HRRPs in the S-band, (**b**) HRRPs in the X-band, (**c**) HRRPs in the Ku-band, (**d**) raw and difference between the two spheres' data.

## 4. Discussions

The difference between the RCS values of the pair of spheres can be used for verifying whether the scattering region is in the same scattering region because the radar calibration should occur in the same scattering region for the calibration object and the calibrated target. First, we can only use the physical cross-sectional area of the sphere to calibrate the radar data in the optic region, and it requires no special terms for the material of the sphere in the optic region. Second, in the Mie region, we should use an object with a similar material composition to the target to calibrate the radar data. In the Mie region, the material composition of the target contributes to the main scattering field. Therefore, we cannot calibrate a target's radar data using a standard object with a different material composition in the Mie region. For example, if the target is a metallic drone, the calibration object could be a metallic sphere. A metallic sphere would not be appropriate if the target was a bird. Since 65% of a bird's mass is water, a water sphere can be used to describe the scattering field of a bird [12]. In this case, water spheres are more appropriate than metallic spheres when calibrating radar echoes from flying birds using S-band weather surveillance radar. Compared to calibration conditions that consider the shape of the standard object, such as rectangular flat plates and cylinders [4], we should pay more attention to the radar bands and the size of the calibration object. In conclusion, having one object in the optic region and the other object in the Mie region would result in inaccurate radar calibration.

The difference between the RCS values of the two spheres also provides a method for ensuring that the test region is in the pure area of the microwave anechoic chamber. Theoretically, the target should be in the far field of the radar scattering field. The distance between the radar antenna and the target must be at least as long as *R*, which is given by [5,9]

$$R > \frac{2d^2}{\lambda} \tag{10}$$

where *d* is the target size in the vertical line of incident direction, and $\lambda$ is the wavelength. In many cases, especially in microwave anechoic chambers, the length of the inside room is shorter than the required range of *R*. For example, in our case, the length of the anechoic section was approximately 5 m. When measuring a large sphere with a diameter of 30 cm in the Ku-band, a range *R* of 9 m is required, which is much longer than the length of the room. Then, how can we ensure that the test area meets the requirements in a compromised way? Or what is the minimum distance between the radar and the target, given the imitated length of the room? The minimum distance is the position where the difference between the two spheres' RCS is equal to the theoretical ones in the optic region. This distance, in our case, is about 2.7 m. Here, we used the Ku-band of the spheres to adjust the antennas and determine the positions of the transmitting antenna and receiving antenna when the

difference between the radar data of the two spheres was approximately equal to the theoretical difference (i.e., 9.5424 dB). Otherwise, the antenna positions may be incorrect, and the test area around the foam holder may not be in the pure area. It requires many attempts to find the distance successfully. First, two spheres with different diameters, such as the large and small spheres used in our test, must be selected. Second, the locations of the antennas must be adjusted to ensure that the difference between the signal amplitudes of the spheres satisfies the theoretical difference of the optic region. The radar bands in the above tests need to be in the optic region for the large and small spheres. In addition to this verification, the other radar bands should be tested. If the location of any antenna changes, the above verification should be repeated. In this kind of anechoic chamber, a small movement of the antenna could cause a considerable change in the signal amplitude of a target. Thus, we need to adjust the positions of the antennas to ensure that the test area is in the pure area. This calibration is conducted using the pair of spheres in the optic region.

The difference between the RCS values of the pair of spheres provides calibrations for the general radar calibration using only one calibration sphere, including (1) verifying whether the scattering regions are in the same scattering region, and (2) checking whether the test region is in the pure area of the microwave anechoic chamber. In addition to the above two benefits brought by the method using a pair of spheres, the locations of the two spheres also provide the location of the target area. However, sometimes in the microwave anechoic chamber, the background clutter is still too high, meaning that the radar signals from a target with a small RCS value may be not distinguished from background clutter. In this situation, the locations of the two spheres can be useful. Taking the results in Figure 4 as an example, both spheres were located at approximately 11.6 m, which indicates the target area's approximate region. Since the distance between the table and the antennas is the same, the drone's radar data and the seagull are also located at approximately 11.6 m. Since the centers of the spheres were at the same location, the difference between the spheres was 10 cm, which is equal to the difference (i.e., 10 cm = 0.15–0.05 m) between the radii of the two spheres. Therefore, this value confirms the location of the target area.

In conclusion, the calibration procedure of the primary measurement setup using a pair of different-sized spheres is described in Figure 5.

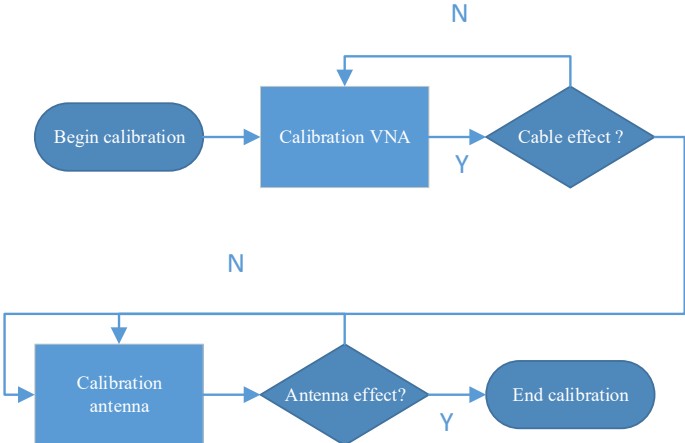

**Figure 5.** Radar calibration procedure using a pair of spheres.

Step (1) Calibrate the VNA to remove the cable effect before connecting the VNA and the antennas. This calibration can be conducted using the standard cable components provided by the VNA.

Step (2) Calibrate the antenna to ensure that the target is in the pure area after connecting the VNA and the antennas with cables.

Step (3) Extract the signals of targets with the range profiles, based on the position located by the spheres, and then process them for the goal of the test.

## 5. Conclusions

Using a standard sphere is a common method for calibrating radar data, but another method is still required to validate the standard sphere. There are at least three questions that must be answered before selecting the calibration sphere. Question 1: How big is the sphere? Question 2: What is the material of the sphere? Question 3: What is the proper distance between the sphere and the antenna? We propose using two different-sized spheres to validate the standard method and provide answers to the above questions. The basic rule is that the calibration should occur in the same scattering region for both the calibrated target and the standard object. The differences in the signal amplitudes of the large sphere and the small sphere can be used to determine the scattering region. Only in the optic region is the difference in the signal amplitudes equal to the theoretical value, and then the size and materials of a standard sphere can vary. Thereby, only when both radar data of the two spheres are in the optic region can their difference in RCS value (i.e., 9.5424 dB) be approximately equal to the theoretical one (i.e., 9.6803 dB). In addition, if the scattering effect is caused by the Mie region, the size and materials of the standard sphere should be comparable to those of the calibrated target; otherwise, the different resonance effects of the target and the sphere could introduce measurement errors. For example, when the calibration is within the S-band, neither the radar data of a large sphere nor of a small sphere can be used for calibrating the radar data of the seagull and the drone within the S-band. In addition, the location of the radar signals of the spheres also provides the location of the target. In brief, the two spheres can both act as calibration objects for each other and other targets and then be used to provide radar calibration.

**Author Contributions:** Conceptualization, J.G. and J.Y.; methodology, J.G.; software, D.K.; validation, J.Y., D.K. and D.L.; formal analysis, J.G. and D.K.; investigation, J.G. and D.K.; resources, D.L.; data curation, D.K.; writing—original draft preparation, J.G.; writing—review and editing, H.H.; visualization, H.H.; supervision, J.Y. and D.K.; project administration, D.L.; funding acquisition, D.K. All authors have read and agreed to the published version of the manuscript.

**Funding:** This research received no external funding.

**Institutional Review Board Statement:** Not applicable.

**Informed Consent Statement:** Not applicable.

**Data Availability Statement:** Not applicable.

**Acknowledgments:** The authors would like to thank the assistants at the Marine Equipment Inspection & Testing Co., in Qingdao, China, including Wenjing Bao and Shangde Wu.

**Conflicts of Interest:** The authors declare no conflict of interest.

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
