# Peer review of "Using a Pair of Different-Sized Spheres to Calibrate Radar Data in the Microwave Anechoic Chamber"

_applsci, doi:10.3390/app12178901_

Round 1

Reviewer 1 Report

The manuscript:

“Using a pair of spheres to calibrate radar data in anechoic chamber” by J. Gong, J. Yan, D. Li, H. Hu. and D. Kong (Ref. No.: applsci-1790232-peer-review-v1)

contains interesting results and may be suitable for publication. However, it is not well-organized and requires a significant elaboration. In particular, the novelty of this work is vague. From the manuscript it is not clear if the similar works has been performed already. Therefore, the authors should stronger emphasized the originality of this study showing its uniqueness.

English is acceptable. However, few minor corrections are required. Furthermore, the manuscript requires some citations.

Apart of this, the following should be also taken into consideration:

Abstract

1) The Abstract should reflect the key quantitative results obtained in this study.

1. Introduction

1) Equation (1) should be cited.

2) The sentence: “For example, the authors reported that they calculated …”, is confusing. The corresponding paper should be cited.

3) The sentence: “In this paper, we propose using a pair of different-sized spheres to verify the standard method using a standard sphere”. What is the objective of this paper? And what are the major results of this study? These questions should be briefly considered in the ending part of the section Introduction.

2. Materials and Methods

1) The sentence: “Fig. 1b shows a photo of the test 80 site, and a pair of antennas can be observed”, is not properly written and should be improved.

2) The sentence: “Theoretically, the RCS values of the two spheres (σ_{a10}, σ_{a30}) in the optic region can be calculated with Formula (2), resulting in values of approximately -21.0491 dBm2 and -11.5067 dBm^2, respectively”. How did you find these theoretical results?

3) Perhaps Fig. 2a should be represented in a conventional way rather than to take a snapshot. The axes and figure captions are not shown in this snapshot.

4) The sentence: “The electric size of a sphere is given by …”. What is “electric size”?

3. Results

1) The sentence: “In brief, resonance effect in the Mie region could be the primary explanation why the RCS values of the two spheres differ from their physical cross-sections”, should be cited to support this claim.

2) The sentence: “The large sphere had higher signal amplitudes in the three radar bands than the small sphere”. What is the ratio of two signal amplitudes?

3) The sentence: “In addition, the position shifts of the small sphere are more likely to be affected by the radar bands than those of the large sphere”, should be cited.

4) The sentence: “The above results explain why the RCS values of the two spheres differ from their geometrical cross-sections because of Mie theory”, should be cited.

4. Discussions

1) The sentence: “Therefore, we cannot calibrate a target's radar data using a standard object with a different material composition”, is confusing and needs clarifications. In particular, it is not clear why a different material composition can affect the calibration of the target’s radar. More description is required.

2) The sentence: “Briefly, having one object in the optic region and the other object in the Mie region would result in inaccurate RCS data”, should be cited.

3) As it has been mentioned above, the novel (original) results showing the uniqueness of this study should be stated.

5. Conclusion

Similar to the Abstract, the section Conclusions should also show the key quantitative results.

The manuscript requires the major mandatory revision.

Reviewer 2 Report

1. Is equation (3)  ?r = ?r − ?t + ?t in dB?

2. Table 1 indicates VNA frequency of 14-18 GHz while it is mentioned in the text for meaurements in S- X- and Ku- bands. It is inconsistent.

3. It is not clear to me that the column 1 (ka) in Table 3 under each band is? It appears is the wavelength then the authors should express the correct (and consistent) units. It is confusing.

4. In the range profile plots, it is clear that the peak of the small sphere arrives later as compared to the large sphere. This is typical as the main reflection point on the spheres will be different due to the difference in radius (0.1m one way). Also the authors bring out three possible reasons for the position shift. They should be carefully checked and present to the readers the cause of the shift, instead of letting the readers to guess.

5. There is strong scattering in range profile plots before the sphere response that occurs around 11.3m that clearly is not related to the sphere. My question is whether background subtraction has been done for these measurements. If it has not been done, the authors should re-do the measurement with background subtraction, as typically been done in RCS measurement, and re-analyze their results.

6. Since the authors have exact RCS data for perfectly conducting spheres,  does it matter what size and material of the sphere is as calibration reference? One can simply measure the RCS of the target and RCS of the perfectly conducting reference sphere and then calculate the RCS data of the target based on the exact RCS data of the sphere? If measurement is done carefully, the measured RCS of the sphere should have similar behavior as the exact sphere (whether it is in Mie region or optical region) and the calibration should be able to determine the RCS of the target under test. If equation (2) is used as the RCS of the reference sphere, then of course there will be some error in the calibration. 

7. If the range of the chamber is not long enough for the far field of any given target, it is intrinsically not suitable for RCS test of the target. The authors discussed in the paper to use the two spheres and adjusting the transmit/receive antennas to "create" a "pure region"? I think this is restricted to very narrow band measurement. In addition, the "pure region" can be due to the sum fields from the transmit antenna with other stray signals from the chamber that can result in false "pure region" for RCS test.

8. Section 4 (Discussions) is very difficult to follow.

Round 2

Reviewer 1 Report

The manuscript: 

“Using a pair of spheres to calibrate radar data in anechoic chamber” by J. Gong, J. Yan, D. Li, H. Hu. and D. Kong (Ref. No.: applsci-1790232-peer-review-v2),

has been improved after major revision. In particular, authors emphasized the objective and novelty of this work. They added additional material, provided clarifications and required citations. However, a few minor amendments to the manuscript would be recommended as follows:

1) The sentence: "... RCS value can be calculated theoretically using different methods in different scattering regions". Again not clear about different method. What are these scattering regions. More information is required.

2) The authors should briefly state about their achievements at the end of Introduction.

3) The sentence: "If we calculate the mean amplitude within three radar bands (i.e., S-band, X-band, Ku-band), the ratio values of that large sphere to the small sphere are 0.5589, 0.5468, and 0.5461, respectively". How did you calculate the mean amplitude values? This should be explained in more details.

4) The sentence: "When any sphere is in the Mie region, the RCS values of the two spheres 224 differ from their geometrical cross-sections". What are these RCS values?

5) Specify the benefits in the sentence: "In addition to the above two benefits brought by the method using a pair of spheres ...".

The manuscript requires a minor revision.
